# Rapid Detection of Thermal Treatment of Honey by Chemometrics-Assisted FTIR Spectroscopy

**DOI:** 10.3390/foods10112892

**Published:** 2021-11-22

**Authors:** Olga Antonova, Javier Calvo, Andreas Seifert

**Affiliations:** 1CIC nanoGUNE BRTA, 20018 San Sebastián, Spain; a.seifert@nanogune.eu; 2CIC biomaGUNE, Basque Research and Technology Alliance (BRTA), 20014 San Sebastián, Spain; jcalvo@cicbiomagune.es; 3IKERBASQUE, Basque Foundation for Science, 48013 Bilbao, Spain

**Keywords:** FTIR, honey quality, PLS, LDA, temperature treatment

## Abstract

Honey, as a nutritious natural sweetener produced by honeybees, offers a unique biochemical composition with great benefit to human health. Transportation and storage conditions as well as violations of processing can lead to decomposition of vitamins, destruction of the integrity of the antioxidant components and enzymes, and further biochemical changes with impact on nutritional quality. We developed a fast detection method of adulterations or changes of honey caused by thermal exposure, which does not require any sample pretreatment. By Fourier-transform infrared spectroscopy, supported by chemometrics methods, we investigated three types of raw honey before and after heat treatment for varying exposure times at different temperatures. Applying principal component analysis and linear discriminant analysis to the preprocessed spectroscopic data, allowed us to discriminate raw honey from thermally altered ones even at low temperatures of 40 °C with high accuracies ≥90%.

## 1. Introduction

Honey is a natural biological high-value product, produced by honeybees from the nectar of blossoms, and collected and combined with specific substances of their own. The main components of honey are sugars—fructose, glucose, maltose, sucrose. Moreover, it contains acids, proteins, minerals, vitamins, phenols, enzymes and other substances, in total more than 400 different substances [1]. There are many different reports of the use of honey from ancient times, not only as food but for medical purposes, and it has found its place in modern medicine too [2,3,4].

Raw honey, which comes “straight from the beehive” without any treatment, contains bee pollen, propolis, and beeswax, which have antibacterial activity and may function as antioxidants, but also some undesirable materials, such as yeast, that are to be removed for better product quality and shelf-life [5,6]. Even though improper processing can be detrimental to the quality of honey and its biological and bioactive chemical properties, there is no guideline for honey treatment based on types and origin.

Commercial processing of honey is usually accompanied by filtering and heating in order to purify, filter, facilitate packaging, to inhibit microorganism growth, reduce moisture content to standard level, and to delay the crystallization process [7,8]. In earlier studies, Fourier-transform infrared (FTIR) spectroscopy was used with chemometrics to categorize honey and detect hydroxymethylfurfural (HMF) level and diastase activity [9,10,11]. Raw honey, containing more than 20% moisture, is easy to ferment, independent of the initial yeast count that affects the honey quality and shelf life [12]. The two most sensitive parameters regarding heating process are HMF content and diastase activity. HMF is an organic compound formed by the dehydration of certain sugars and various processed foods in acidic environments when they are heated through the Maillard reaction [13]. HMF content shows high values in honey that has undergone heat treatment, stored in non-adequate conditions, or adulterated with invert syrup [14]. Based on the European Honey Directive, HMF content is used for quality control of honey and should not exceed 40 mg/kg, excluding honey of declared origin from regions with tropical climate and blends of these honeys, for which the limit is 80 mg/kg (Council Directive 2001/110/EC).

To reduce the water content in honey below 20% and eliminate yeast destruction for shelf-life prolongation, a wide range of heating temperatures–ranging from 30 to 140 °C is applied by honey producers worldwide, with treatment time ranging from a few seconds up to several days. It was investigated that heating up to 80 °C, between 60 and 30 s destroys all microorganisms responsible for quality damage without spoiling the honey [15]. This treatment was considered a mild or more suitable treatment condition. Results of the investigation of honey from Crete [16] showed a significant alteration of quality parameters under heating at 65 °C for 6 h. Pine honey was the most resistant sample to HMF formation in all heating procedures, and multifloral honey was the least altered in its enzymatic activity through the whole thermal process. As a result, optimal heating conditions strongly rely on geographical and botanical origin of honey, however, in general, heat processing should be reduced where possible to maintain health-promoting effects.

To preserve nutrients in honey as long as possible, its moisture content should be less than 17.1% and storage temperature below 11 °C [8]. However, it is not always possible to follow treatment and storage requirements due to weather conditions during the extraction of honey or during transportation. As quality of honey in terms of bioactivity is important, it is the end-user’s right to receive controlled or certified honey that proves absence of falsification or improper storage.

Taking such requirements and boundary conditions into consideration, the aim of this study was to develop a protocol for rapid detection of possible heat treatment of raw honey, subject to transportation, storage, or even adulteration. The developed method is based on chemometrics-supported spectroscopy. Using FTIR spectroscopy, we investigated the spectra of three types of honey, both raw and thermally treated, with varying time intervals and temperatures for the treatment. As chemometric methods, we employed PCA (principal component analysis) and LDA (linear discriminant analysis) for discrimination and multinomial classification and confirmed our results by chromatography as reference method.

## 2. Materials and Methods

Commercially available raw honeys from eucalyptus, acacia, and orange blossom were analyzed in this study (Bona Miel organic, Alicante, Spain). According to the declaration of the manufacturers on the labels, honey was harvested both in the European Community (EC) and non-EC countries. Raw honey samples were supplied directly as they had been obtained from beekeepers and had not undergone any heating or filtration. Before the analysis, the samples were stored in a dark place at room temperature. From each honey type, we took three samples and performed three measurements each; corresponding results are given in Table 1. The moisture of honey was detected by refractometry, using the Handheld Digital Refractometer PCE-DRH 1 Honey (PCE Instruments UK Ltd., Southampton, United-Kingdom). Pre-treatment was done in a water bath at 40 °C until visible dissolution of any crystallization. The moisture content of honey samples was measured at 20 °C.

pH was determined according to the method described in the “Techniques for the Evaluation of Physicochemical Quality and Bioactive Compounds in Honey” [17]. Ten grams of honey were dissolved in 75 mL of distilled water; thereafter a direct reading was taken for each honey sample from a pH meter (VWR pH100, YSI Inc., Yellow Springs, OH, USA) calibrated with appropriate buffers of pH 4.0 and 7.0.

Sugar content was determined by GC-FID (gas chromatography, flame ionization detector; Agilent, Santa Clara, CA, USA) following the described Pierce–Portallier method [18]. For the GC-FID analysis, Openlab CDS Chemstation software was used (Agilent, Santa Clara, CA, USA). Briefly, calibration standard (glucose, fructose, and sucrose) and honey samples (50 mg/mL) were prepared in miliQ water containing 10 mg/mL of fucose as internal standard. Then, 30 μL of samples were dried at 50 °C overnight and 200 μL oxime reagent was added to dried samples, mixed and heated at 70 °C for 30 min. Next, samples were cooled to room temperature and 100 μL of hexamethyldisilazane and 10 μL of TFA (trifluoroacetic acid) were added and mixed for 30 s. Samples were centrifuged for 30 s and the supernatants were placed on GC vials. The quantification of sugars was performed by an 8890 Agilent GC-FID system (Agilent, Santa Clara, CA, USA). The separation was carried out using an Agilent HP 5 column (30 m, 0.32 mm, 0.25 µm) (Agilent, Santa Clara, CA, USA), the injection volume was 1 μL and the injector temperature was set to 280 °C. The oven temperature was initially set to 100 °C and was increased for 17 min at 10 °C/min until 270 °C. Finally, the temperature was held at 20 °C for 5 min. Total run time was 22 min.

HMF was quantified by an ACQUITY UHPLC (ultra-high-performance liquid chromatography) system equipped with a photodiode-array detector (Waters, Mildford, MA, USA). For the UHPLC data acquisition and analysis, Masslynx software v4.1 was used (Waters, Milford, MA, USA). Honey samples (50 mg/mL of honey in milliQ water) and HMF calibration standards were prepared immediately before their analysis. The separation of HMF was performed using an Acquity BEH C18 reverse phase column (50 × 2.1 mm, 1.7 μm) (Waters, Mildford, MA, USA) at 30 °C. The elution buffers were 0.1% formic acid in water (A) and acetonitrile (B), and the chromatographic method was run under the following gradient conditions: 99% A over 1 min, 99–1% over 1 to 6 min, 1% for 2 min, 1–99% for 0.5 min and 99% A for 1.5 min before the next injection. The column temperature was set at 30 °C, the injection volume was 5 μL, and the flow rate was kept constant at 300 μL/min. The detection and quantification of HMF was obtained after monitoring the UV signal at 284 nm wavelength.

FTIR measurements for honey did not require any sample preparation; a single droplet was put onto the ATR crystal. FTIR spectra were taken with the Frontier MIR (mid-infrared) Spectrometer (Perkin Elmer, Waltham, MA, USA), applying the attenuated total reflectance (ATR) technique. Spectra were recorded with 48 scans per spectrum from 4000 to 600 cm^−1^. PerkinElmer software (Perkin Elmer, Waltham, MA, USA) was used for the FTIR data acquisition. To minimize the influence of temporal baseline shifts, a background spectrum against air was recorded before each sample spectrum. In order to prevent influence from measurement artifacts on a longer timescale, the spectra were measured in randomized order. To improve the signal-to-noise ratio, a total number of 50 spectra from each sample was acquired and averaged for further analysis; technically, only one drop of honey was required for each spectrum to cover the ATR crystal. For the combined method used, consisting of the FTIR spectroscopy unit, data preprocessing, and machine learning algorithms PCA and LDA, and with its inherent resolution and uncertainties, we observed a kind of threshold characteristic for the discrimination of time-dependent heating. At 40 °C thermal treatment, only after around 3.5 h a clear discrimination from raw honey could be identified, and for the case of 70 °C, the observed threshold for clear discrimination starts after 15 min.

Temperature and time of thermal treatment were based on common heating practices by beekeepers and manufacturers during honey processing and average time of the delivery process [19]. Moreover, honey heat-treatment at temperature lower than 75 °C does not lead to fast HMF formation [20]. Equal volumes of each honey were placed in 5 mL Eppendorf vials; thermal treatment was carried out in the Thermomixer Comfort (Eppendorf AG, Hamburg, Germany) at 40 °C for 3.5, 5.5, 7.5, and 24 h and at 70 °C for 15, 30, 60, 90, and 120 min; honey was cooled down to room temperature in a dark place for 4 h before further analysis. Because of slow mutual transformation processes between sugars that influence the clear separation of honey that underwent thermal treatment, we present in the main body of the paper data with the extreme values of thermal treatment, at 70 °C for 15 and 120 min and at 40 °C for 3.5 and 24 h. Accordingly, the results give a clearer picture of our findings. The complete data with all heating periods is shown in the Appendix A.

Before analyzing the measured spectra by chemometric methods, and to guarantee sample heterogeneity for obtaining reliable results, the data were preprocessed by the following steps. Due to changes in the dynamics of the spectra, best results were obtained using min-max normalization (numpy 1.18.1 “min” and “max” functions) for all FTIR spectra and then smoothed using a Savitzky–Golay filter with 5 cm^−1^ window width and third-order polynomials. PCA and LDA were realized for raw and heated honey samples using the Python-based libraries Numpy [21] and Scikit-learn [22]. PCA is a statistical method for reducing the dimensionality of datasets, but at the same time minimizing information losses [23]. LDA models the differences in the data, in a way that the differences in intraclass variations is maximized. Calibration and validation models were developed to predict heating treatment on the basis of the spectral information mainly in the region of 800–1500 cm^−1^. This spectral region contains all necessary characteristic bands related with sugar transformations. Any other spectral regions introduce more noise into the data and downgrade the results. For cross-validation of our data, we used one of the common techniques, the train-test-split approach, which randomly splits the complete data into a training set of 75% and a test set of 25% of the data; then we applied LDA.

## 3. Results and Discussion

FTIR spectra of both raw and thermally treated honey show various narrow vibrational bands in the spectral range from 600–4000 cm^−1^, as shown in Figure 1a. Characteristic bands in the FTIR spectra of honey related to the content of carboxylic acids, alcohols, carbon, and aromatic C–H groups are presented in Table 2, based on prior assignments by Kedzierska-Matysek et al. [24] and Kasprzyk et al. [25].

By visual inspection, it is difficult, or rather impossible, to identify the FTIR signals responsible for the changes during heat treatment because of the complex spectra and very minor changes. To unveil hidden information from the spectra, different kinds of chemometrics analysis were used to process the spectral data and to determine the difference between raw and heated honey.

Spectral data span a high-dimensional space and cannot be used for classification via discriminant analysis without preprocessing. PCA, as one of the most used multivariate analysis techniques, was employed to transform the data set into a reduced new set of variables. We found that a full spectral range without preselection did not lead to an acceptable result. Few works have reported on feature extraction from FTIR spectra, such as the definite wavelength range or using derivatives [26]. In our case, using the spectral region 800–1500 cm^−1^ with the main characteristic bands, spectra have been classified correctly with high accuracy.

The variability of the PCA result for the dataset projected onto the first two principal components (PCs) is shown in Figure 1b. PCA is not a quantitative classifier, however, it constitutes a useful tool for the first visual inspection. Moreover, PCA is the most promising tool to detect the adulteration of honey with respect to different adulterant sugars: glucose, fructose, sucrose, etc. [11,27]. From the figure it is evident that the composition of each honey is very different, moreover, the composition of eucalyptus honey changes a lot during the heating. For a more detailed analysis, each honey was investigated separately.

Eucalyptus honey was heated at 70 °C for 15, 30, 60, 90, and 120 min. The maximum changes during heating occurred in the range of 940–1100 cm^−1^. As shown in Figure 2, temperature increasing leads to gradual changes in the ratio between different bands. It is obvious from FTIR spectra that honey heating leads to a change in the ratio of the intensities of the bands at 990 cm^−1^ and 1050 cm^−1^, corresponding to fructose and glucose respectively [28]. These changes could be attributed either to Maillard reaction or sugar intermutations. Since Maillard reaction is pH-dependent (due to pH-dependent protonation of amino acids), different honeys have different reaction rates, and as a consequence, will require individual optimization of the model for exact honey type and species. 

Figure 3 shows how PCA, applied to FTIR spectra, can separate raw and treated eucalyptus honey. Even though thermal treatment for 15 and 120 min at 70 °C cannot be separated completely by PCA, we can see a clear differentiation. Based on loadings of PCs, the major distinctive features between raw and heated honey are located at ~990 cm^−1^ and 1050 cm^−1^, which coincides with the results obtained from spectra before.

To further validate, better classify, and finally quantify the three different classes of samples, we applied LDA as a classifier. Figure 4 demonstrates that the developed classification model, based on a training set of 38 samples, is nicely verified by the test set, consisting of 12 samples for raw honey. In comparison with the PCA score plot, LDA is able to clearly distinguish between the honey samples that had been treated for 15 and 120 min at 70 °C, with test sets of 11 and 12 samples, respectively.

As a validation method, a confusion matrix was calculated, whose corresponding results are listed in Table 3. As major figures of merit, accuracy, precision, recall, and F1 score have been calculated. Overall accuracy represents an average performance of multiclass results, whereas precision, recall, and F1 score correspond to the performance of each class.

Out of 38 honey samples in these three different groups, 36 samples were correctly classified, yielding an overall accuracy of the model of 0.947.

In a second experiment, the thermal treatment was done at 40 °C at different heating times according to the explanations in the Section 2. Again, PCA delivers a well-clustered visual discrimination in a two-dimensional visualization of raw honey from honey after thermal treatment. However, as before, the two differently treated honey samples cannot entirely be separated (see Appendix A). As in the previous case, the dominant spectral feature for differentiation is located at 987 cm^−1^.

Again, we applied LDA for classification, with the corresponding graphical illustration in Figure 4b. A decrease of heating temperature led to a decrease of the overall classification accuracy; however, raw honey was still clearly differentiated from thermally treated honey with an accuracy of 100%. The outcome of the confusion matrix is listed in Table 4.

One of the most sensitive parameters regarding heating process or storage in non-adequate conditions is the HMF content. However, based on literature research, FTIR is not sensitive enough for detecting changes less than 13 mg/kg [26]. Additionally, Tosi et al. observed that HMF content didn’t exceed the 40 mg/kg acceptance limit even for heating at 140 °C for 60 min [15]. Thereby changes in HMF during thermal treatment do not deliver significant input that is sensitive enough for FTIR spectroscopy, which was confirmed by UHPLC/UV measurements (Appendix A).

Another parameter sensitive to heating is the carbohydrate content. A characteristic feature of honey is its composition. That depends on floral source, geographical origin, seasonal, and environmental factors. Accordingly, this greatly complicates the quality control of honey. To confirm that our PCA/LDA model, mostly based on glucose and fructose concentration, works correctly, the relative changes of fructose and glucose concentration during the heating were calculated based on the data of chromatography, taking total concentration of sugars as 100% (Appendix A). Figure 5 confirms, that the higher the changes in sugar concentrations are due to heating, the more accurately the model works. These changes are related to the composition of the honey and its pH. Additionally, Appendix A shows that the major changes during heating occur in the first few minutes of heating, independent of temperature.

It turns out that a special strength of our model is its ability to determine even small mutual transformations in sugar composition that occur during heating. Such transformations could also be revealed by chromatography; however, this technique requires much more complex sample preparation and takes much more time than chemometric analysis. The complete results with data for accuracy and precision for acacia and orange blossom honey are given in the Appendix A. These results clearly indicate that the model works independently of honey type and composition.

## 4. Conclusions

The combination of Fourier-transform infrared spectroscopy with chemometric methods proves a powerful technique for quick and high accurate evaluation of the quality of honey. For FTIR analysis, no specific sample preparation is required. For three different types of honey, it has been shown that by the combination of said methods, thermal treatment of honey can be reliably detected. Heating of honey can generally have two origins, first, it can happen during transportation or storage, and, second, by intended heating regardless of the purpose. Principal component analysis (PCA) demonstrates the capability of differentiating between thermally treated and raw honey. Beyond that, by linear discriminant analysis (LDA) we can also quantitatively discriminate different conditions of thermal treatment.

In summary, FTIR spectroscopy complemented by chemometric methods allows for a quick and easy control of the quality of honey and can give a precise indication of unfavorable transport or storage conditions, or adulteration in case of incorrectly labeled raw honey.

## Figures and Tables

**Figure 1 foods-10-02892-f001:**
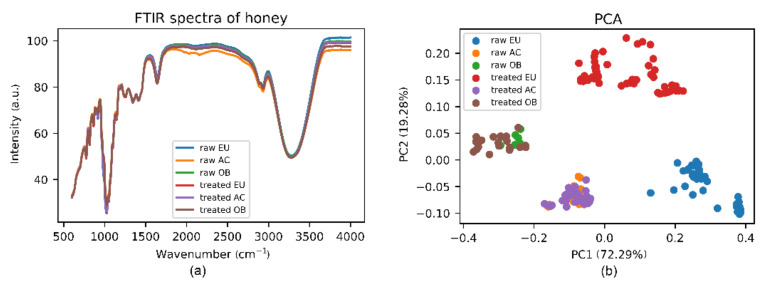
Raw and thermally treated honey samples—120 min at 70 °C. (**a**) FTIR spectra. (**b**) Principal component score plot based on PCA (principal component analysis) applied to the spectral range of 600–4000 cm^−1^ from FTIR data. EU: eucalyptus, AC: acacia, OB: orange blossom.

**Figure 2 foods-10-02892-f002:**
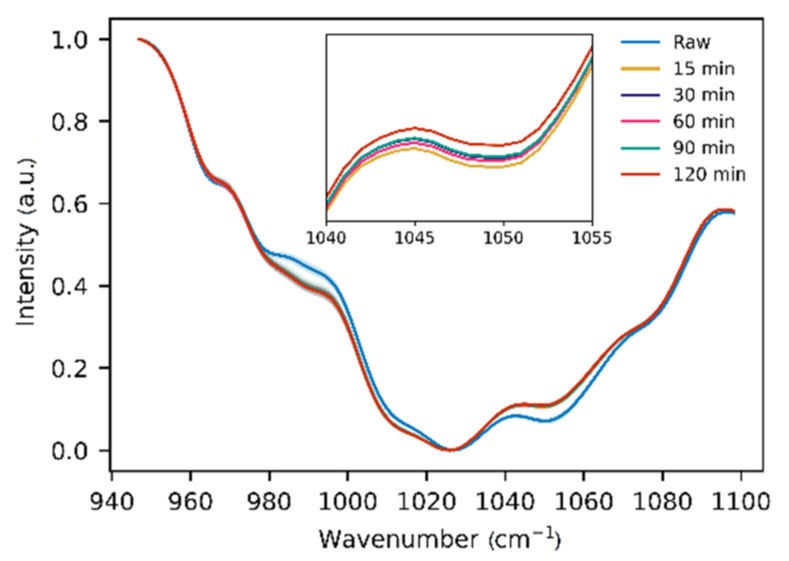
Fingerprint region of FTIR spectra of eucalyptus honey before and after thermal treatment at 70 °C for 15, 30, 60, 90, and 120 min.

**Figure 3 foods-10-02892-f003:**
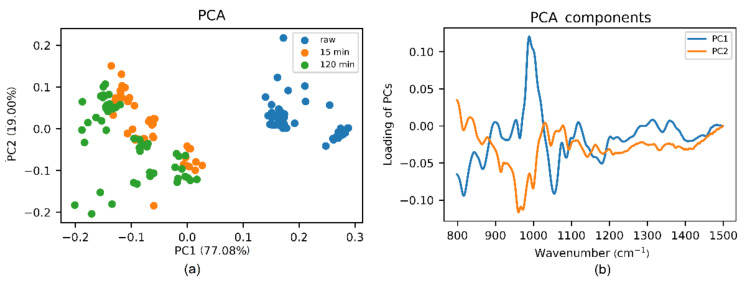
(**a**) Principal component score plot and (**b**) loading plot of PCs of raw and thermally treated eucalyptus honey samples—15 and 120 min at 70 °C—based on PCA applied to the spectral range of 800–1500 cm^−1^ from FTIR data.

**Figure 4 foods-10-02892-f004:**
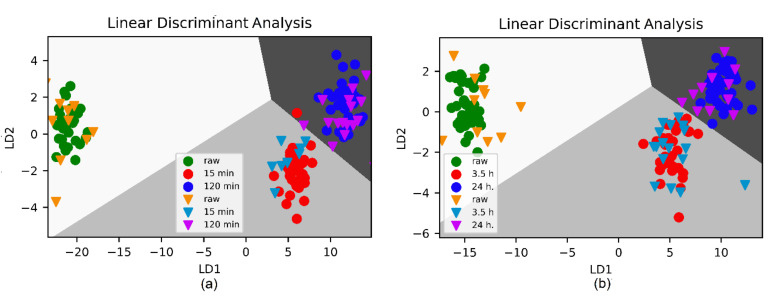
Discrimination of eucalyptus honey obtained by LDA of FTIR spectra based on analysis of the spectral range of 800–1500 cm^−1^. (**a**) Samples heated at 70 °C. Achieved overall accuracy is 0.947. (**b**) Samples heated at 40 °C. Achieved overall accuracy is 0.895. Dots: training set, triangles: test set. LD – linear discriminant.

**Figure 5 foods-10-02892-f005:**
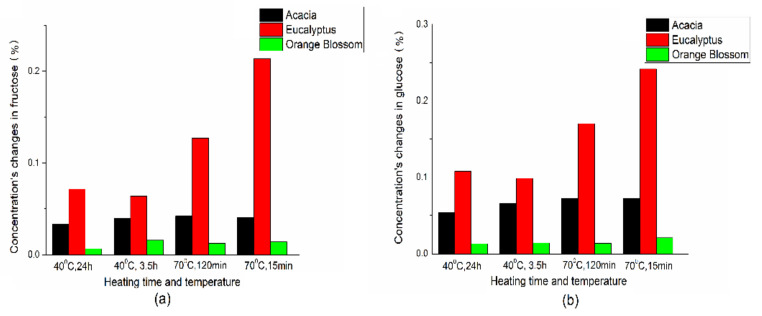
Relative changes in fructose (**a**) and glucose (**b**) concentrations, due to heating at different time and temperatures based on chromatography data (Appendix A).

**Table 1 foods-10-02892-t001:** Physicochemical parameters including standard deviation (SD) of different raw honey types. Data are based on three samples per honey type and three measurements each.

Honey Type	Moisture (%)	Fructose (%)	Glucose (%)	Sucrose (%)	pH	HMF (mg/kg)
Eucalyptus	18.21 ± 0.16	38.5 ± 1.2	21.33 ± 0.66	0.136 ± 0.012	3.81 ± 0.12	26.51 ± 0.87
Acacia	17.54 ± 0.13	32.45 ± 0.14	20.9 ± 3.0	0.3251 ± 0.0031	4.15 ± 0.14	9.43 ± 0.53
Orange blossom	17.53 ± 0.12	34.36 ± 0.19	24.71 ± 0.62	4.078 ± 0.018	3.89 ± 0.11	16.10 ± 0.64

HMF: hydroxymethylfurfural.

**Table 2 foods-10-02892-t002:** Assignment of absorption peaks of the FTIR spectra from honey.

FTIR Wavenumber (cm^−1^)	Type and Origin of Vibration
3281	υ(O-H) in H_2_O
2933, 2890	υ(C-H) or/and υ(NH_3_) of free amino acids
1639	σ(OH) from H_2_O
1456, shoulder	δ(O-CH) and δ(C-C-H)
1415	δ(O-H) in C-OH group + δ(C-H) in alkenes
1342	δ(OH) in C-OH group
1249, 1189, 1148	υ(C-H) in carbohydrates or/and υ(C-O) in carbohydrates
1100	υ(C-O) in C-O-C group
1051, 1023, 981, 965	υ(C-O) in C-OH group or υ(C-C) in the carbohydrate structure
919	δ(C-H)
894, 865, 817	anomeric region of carbohydrates or δ(C-H)

ν—stretching vibrations, δ—deformation vibrations.

**Table 3 foods-10-02892-t003:** Precision, recall, F1 score, and overall accuracy for raw and heated eucalyptus honey at 70 °C for different heating times.

Group	Precision	Recall	F1 Score	Overall Accuracy
Raw	1.000	1.000	1.000	0.947
Heated 15 min	1.000	0.818	0.900	
Heated 120 min	0.875	1.000	0.933	

**Table 4 foods-10-02892-t004:** Precision, recall, F1 score, and overall accuracy for raw and heated eucalyptus honey at 40 °C for different heating periods.

Group	Precision	Recall	F1 Score	Overall Accuracy
Raw	1000	1.000	1.000	0.895
Heated 3.5 h	0.765	1.000	0.867	
Heated 24 h	1.000	0.667	0.800	

## Data Availability

The data presented in this study are available within the manuscript and the Appendix A.

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
