# Peer review of "Rapid Detection of Thermal Treatment of Honey by Chemometrics-Assisted FTIR Spectroscopy"

_foods, 2021, doi:10.3390/foods10112892_

Round 1

Reviewer 1 Report

Detecting adulteration in honey is a very important point in food analysis. Therefore, these studies are very important.

The publication requires minor corrections:

  1. line 48 - exceptions should be added - when higher HMF levels are legally allowed
  2. line 146 - Fig A1 and A2 are given while Supplementary Information are given Fig S1 and S2
  3. line 163 - the name of the author of the publication was entered incorrectly 
  4. Why has eucaliprus honey been discussed extensively, whereas acacia and orange honey are not
  5. Reference - adapt references to the journal's requirements

Author Response

Dear review, 

Reviewer 2 Report

The manuscript under appreciation is about the detection of thermal treatment of honey by the employment of FTIR spectroscopy in combination with principal component analysis and linear discriminant analysis.

The manuscript is interesting and provides novelty however, there are several issues:

In the “Introduction” section the authors should report whether the thermal treatment of honey has been previously studied by other researchers by FTIR spectroscopy.

In the “Materials and Methods” section, line 78, the authors must provide the exact number of samples from each honey type.

Regarding Table 1:  How many samples did the authors use from every honey type for the analysis of physicochemical parameters? If they have used more than one sample, then the mean values along with standard deviation must be reported. In the first row of the table the word “Honey.” should be changed to “Honey type”. Furthermore, the authors should clarify in the text how many replicates they performed for each of the physicochemical parameters. The number of significant figures is indicative of the repeatability of the measurement, please report how many times did you performed the analysis and correct accordingly (triplicates?).

For the GC-FID (line 95), UHPLC (line 108), and FTIR (line 119) the authors must provide the data acquisition software.

In line 126 the authors report that 50 replicates for each sample were acquired. I suppose that afterward, the authors used the average spectrum from each sample for the statistical analysis. Please clarify this in the main text.

Line 149: why did the authors choose to perform min-max normalization against other normalization methods? Which software was used for the normalization and smoothing of spectral data? Please clarify in the text.

Lines 156-157: Why did the authors choose the spectral region of 900-1500 cm-1 for the statistical analysis?

In the legend of Figure 1, the PCA was applied in 800-1500 cm-1, it is not consistent with the spectral region reported in line 156 (900-1500 cm-1).

Table 2, row 1: In the “cm-1” the “-1” should be superscript. In the 5th row, the wavenumber “145” is mistaken, please correct.

In order to apply LDA, the Box M test (Equivalence of Covariance Matrices) must be non-significant. Did the authors perform the  Box M test? Please add this information to the manuscript. Furthermore, the authors should provide the Eigenvalues and Wilks lambda values from the LDA for each discriminant function. The LDA model must be validated either using cross-validation or external validation. The authors must provide validation data.

Author Response

Dear review, 

Round 2

Reviewer 2 Report

The authors addressed all the issues and the manuscript has been significantly improved. The manuscript is suitable for publication.

Author Response

Dear review, 

We followed the recommendation of improving the English language by a native speaker.

With best regards,
Olga Antonova and Andreas Seifert